# Personalized computational model quantifies heterogeneity in postprandial responses to oral glucose challenge

Balázs Erdős[1,2]⊛ *, Bart van Sloun[1,2]⊛, Michiel E. Adriaens[1,2], Shauna D. O'Donovan[3], Dominique Langin[4], Arne Astrup[5], Ellen E. Blaak[1,6], Ilja C. W. Arts[1,2‡], Natal A. W. van Riel[7‡]

**1** TiFN, Wageningen, The Netherlands, **2** Maastricht Centre for Systems Biology (MaCSBio), Maastricht University, Maastricht, The Netherlands, **3** Division of Human Nutrition and Health, Wageningen University, Wageningen, The Netherlands, **4** Institut National de la Santé et de la Recherche Médicale (INSERM), Université Paul Sabatier Toulouse III, UMR1048, Institute of Metabolic and Cardiovascular Diseases, Laboratoire de Biochimie, CHU Toulouse, Toulouse, France, **5** Department of Nutrition, Exercise and Sports, Faculty of Science, University of Copenhagen, Copenhagen, Denmark, **6** Department of Human Biology, NUTRIM School of Nutrition and Translational Research in Metabolism, Maastricht University, Maastricht, The Netherlands, **7** Department of Biomedical Engineering, Eindhoven University of Technology, Eindhoven, The Netherlands

⊛ These authors contributed equally to this work.
‡ These authors also contributed equally to this work.
* balazs.erdos@maastrichtuniversity.nl

**Data Availability Statement:** The data supporting the findings in this study are available to eligible researchers from Gabby Hul

## Abstract

Plasma glucose and insulin responses following an oral glucose challenge are representative of glucose tolerance and insulin resistance, key indicators of type 2 diabetes mellitus pathophysiology. A large heterogeneity in individuals' challenge test responses has been shown to underlie the effectiveness of lifestyle intervention. Currently, this heterogeneity is overlooked due to a lack of methods to quantify the interconnected dynamics in the glucose and insulin time-courses. Here, a physiology-based mathematical model of the human glucose-insulin system is personalized to elucidate the heterogeneity in individuals' responses using a large population of overweight/obese individuals (n = 738) from the DIOGenes study. The personalized models are derived from population level models through a systematic parameter selection pipeline that may be generalized to other biological systems. The resulting personalized models showed a 4-5 fold decrease in discrepancy between measurements and model simulation compared to population level. The estimated model parameters capture relevant features of individuals' metabolic health such as gastric emptying, endogenous insulin secretion and insulin dependent glucose disposal into tissues, with the latter also showing a significant association with the Insulinogenic index and the Matsuda insulin sensitivity index, respectively.

## Author summary

The postprandial glucose and insulin responses to an identical meal can vary greatly across individuals. Certain dynamic features of these response curves have been shown to

[g.hul@maastrichtuniversity.nl]. The data are not publicly available due to ethical restrictions and privacy of participant data.

**Funding:** The project is organized by and executed under the auspices of TiFN [https://www.tifn.nl/], a public - private partnership on precompetitive research in food and nutrition. Funding for this research was obtained by E.B., M.A., I.A. and N.vR. from DSM Nutritional Products [https://www.dsm.com], FrieslandCampina [https://www.frieslandcampina.com/], Danone Nutricia Research [https://www.nutriciaresearch.com/] and the Topsector Agri&Food [https://topsectoragrifood.nl/] (grant number: TiFN 16NH04). Additional funding by the Netherlands Organisation for Scientific Research [https://www.nwo.nl/] (grant number: ALWTF.2016.021) was awarded to I.A and N.vR. The funders had no role in study design, data collection and analysis, decision to publish, or preparation of the manuscript.

**Competing interests:** The authors have declared that no competing interests exist.

be indicative of the state of the glucose homeostasis and therefore relevant for targeted lifestyle intervention. In this study, we implement personalized computational models of the insulin regulated glucose homeostasis that take advantage of the complete time-courses of postprandial glucose and insulin trajectories following an oral glucose challenge in a large population of overweight/obese individuals. We show that the personalized models capture the responses more accurately compared to population-level models. In addition, the physiological basis of the models provides insight into the variability in the glucose homeostasis of individuals. The model parameters represent relevant features of the metabolic health such as insulin secretion or insulin mediated disposal of glucose into tissues that also correlated with independent measures of insulin secretion and whole-body insulin resistance, respectively. Furthermore, the models can be quantified from comparatively non-invasive sampling techniques and may be readily transferable to the clinic.

## Introduction

Glucose homeostasis is maintained by the complex interplay between several tissues and organs including the brain, pancreas, skeletal muscle, liver and adipose tissue. The evolution of plasma glucose and insulin concentrations during an oral glucose tolerance test (OGTT) provide a readout of the metabolic health of these underlying tissues capturing the effects of insulin sensitivity and metabolic resilience status [1]. Certain features of the standard 75g OGTT response curves are widely accepted to be representative of glycemic control, most notably the fasting and 2 hour post-load glucose values are used in the diagnosis of prediabetes and type 2 diabetes mellitus (T2DM) [2]. The area under the plasma glucose/insulin response curve (AUC) is an extensively employed measure to compare responses [3] and has been successfully used in targeted and even personalized nutrition approaches [4–6]. However, the AUC is a somewhat crude measure that may often lead to ambiguous classifications [7]. Therefore, certain dynamic properties of the glucose response curves e.g. peak time, have been nominated as relevant for pathophysiological characterization [6, 8–11]. In addition, the post-load glucose and insulin trajectories may be used to derive proxy measures of whole-body and tissue-specific insulin sensitivity to serve as a surrogate to the hyperinsulinemic-euglycemic clamp. The HOMA-IR [12] and Matsuda insulin sensitivity indices [13, 14] have been widely utilized to quantify whole-body insulin resistance from fasting and average postprandial glucose/insulin levels, respectively. In recent years, the increased recognition of tissue-specific insulin resistance [15] leading to metabolically distinct phenotypes, has resulted in the development of the HIRI and MISI indices, quantifying hepatic and skeletal muscle insulin resistance from OGTT responses [16–18]. While these measures capture certain aspects of metabolic resilience, they rely upon single time-point or average glucose and insulin values taken from the response curves, as a result the dynamics of the time-courses are largely disregarded. Recently, Hulman et al. have shown that using a latent class mixed models framework, the glucose trajectories of healthy individuals following an OGTT may be classified into four distinct insulin sensitive phenotypes [19]. This approach—making use of the complete time-courses—highlights the importance of the dynamics of the glucose responses, however it does not allow for an individualized exploration due to the limited number of prospective classes.

The move towards personalized interventions requires the characterization of the large heterogeneity in individuals' glycemic regulation. Therefore, a holistic approach, accounting for the dynamic properties of the response curves is needed on the individual level. Furthermore,

the close interplay between plasma glucose and insulin concentrations calls for the evaluation of glucose and insulin trajectories as a whole, rather than as disjoint indicators.

Physiology-based mathematical models of the human glucose-insulin regulatory system can provide quantitative information on the dynamics while capturing the mechanistic link between glucose and insulin. Such models are built to describe the physiological processes by which insulin regulates glucose levels using *a priori* understanding of the underlying biological system. The detail to which the model can accurately simulate the glucose-insulin response mechanism depends on the desired scope and the availability of quantitative data. The Bergman model, a simplistic model of glucose disappearance containing only 5 parameters, has been extensively used to approximate insulin sensitivity and $\beta$-cell function using plasma glucose and insulin values following a frequently sampled intravenous glucose tolerance test [20]. The integrated glucose-insulin model has been used to describe population as well as individual responses to a frequently sampled OGTT [21], however its applicability (to nutritional and metabolic studies) is limited due to the complexity in the model's glucose absorption term that is made possible by an unusually frequent sampling strategy. A more complex model built by Dalla-Man et al. provides a detailed account of the underlying processes governing glucose utilization following a meal [22]. Here, the complexity of the model is enabled by the availability of triple tracer glucose data, quantifying the glucose fluxes between tissues. While the Bergman model can be applied to individual data, the Dalla-Man model has mostly been applied to population average data for *in silico* simulation and testing of insulin pump systems. The Eindhoven-Diabetes Education Simulator (E-DES) is a comparatively simple multi-compartmental model containing 12 parameters that has been used to describe the dynamics of the glucose homeostasis in healthy, type 1 and type 2 diabetic populations [23, 24].

Quantifying uncertainty in model parameters is essential to understand the limitations and predictive power of the model [25]. It is particularly important to consider parameter identifiability when estimating model parameters on the individual level—where sensitivity to measurement error may be high—to retain parameters that can be reliably estimated. Identifiability analysis may be carried out through methods such as Profile Likelihood Analysis (PLA) to evaluate how well parameter values can be determined given the available data [26, 27].

The aim of the current work is to explore the heterogeneity in the glucose and insulin responses to an OGTT in a large population of individuals by developing personalized dynamic models of the insulin mediated glucose metabolism, using an adapted E-DES model. The model parameters are estimated from measured postprandial trajectories of both glucose and insulin, and represent physiologically relevant properties that in turn may be used in the early identification of deterioration in the glucose homeostasis. Furthermore, the workflow presented here for transitioning a dynamic model away from describing population averages and towards individual response patterns may prove useful in numerous other applications, as it is generalizable to other biological models and systems.

## Materials and methods

### Ethics statement

The Medical Ethical Committees of the respective countries approved the DIOGenes study protocol. Participants provided informed written consent, and all procedures were conducted in accordance with the Declaration of Helsinki. Trial registration number: NCT00390637.

### Data

Data from the DIOGenes study (NCT00390637), a pan-European, multi-center, randomized controlled dietary intervention study were used in this work [28]. At the baseline of the

intervention, following an overnight fast (n = 1118) overweight/obese ($BMI > 27kg/m^2$) but otherwise healthy adult participants underwent a two hour 75g OGTT, with plasma samples taken at the fasting state (t = 0) and 30, 60, 90 and 120 minutes after the glucose ingestion. The plasma samples were subsequently analyzed for glucose and insulin concentrations. Responses at the baseline of the intervention were used. Individuals with an incomplete set of glucose and/or insulin measurements were excluded from the analysis.

## Adapted E-DES model

The Eindhoven-Diabetes Education Simulator is a physiology-based mathematical model of the human insulin mediated glucose regulatory system in healthy, type 1 diabetes, and T2DM phenotypes [24]. The two compartment model describes the following physiological processes through coupled differential equations (see model schematic and details in S1 Fig, S1 and S2 Appendices): Glucose mass is emptied into the gut according to an exponential decay function, followed by uptake into the plasma proportionally to the amount of glucose present in the gut. Both glucose and insulin fluxes are considered in the plasma compartment. Insulin secretion from the pancreas is modelled through a proportional-integral-derivative (PID) controller, responding to elevated plasma glucose levels. The insulin response facilitates the insulin-dependent glucose disposal to tissues such as the muscle. In addition, there is a constant glucose removal from the plasma by obligate glucose oxidizers such as the brain or the red blood cells. While the plasma glucose levels are elevated, endogenous glucose production (EGP) in the liver is suppressed. Finally, insulin is cleared by the liver proportionally to the plasma insulin concentration, as well as by a transfer and degradation in the interstitial fluid. The parameters corresponding to these physiological parameters control the rate of change in glucose or insulin concentrations. Through modulation of the parameters, responses of metabolically different phenotypes may be simulated *in silico*. The model has been previously parameterized and validated on multiple OGTT data sets from healthy populations [24]. The E-DES model was implemented and analyzed in MATLAB 2018b (The Mathworks, Inc., Natick, Massachusetts, United States). For the current study population, an adapted E-DES model is used. Model equations, including a description of parameters and modifications are described in detail in S1 Appendix.

## Parameter estimation

Parameters were estimated through minimizing the combined sum of squared residual (SSR) in the model prediction for glucose and insulin (Eq 1) using *lsqnonlin*, a non-linear least squares solver in MATLAB. To avoid becoming trapped in erroneous local minima, the optimal parameter sets were obtained following fifty initializations of the optimization algorithm with 25% random noise starting from the original parameter value for the average healthy population from the publication [24].

$$SSR = \sum_{j=1}^{m}\sum_{i=1}^{N}(\gamma((y_{i,j}|\vec{\theta}) - d_{i,j}))^2 \tag{1}$$

Where m, and N represent the number of metabolites and the number of time-points, respectively. The measured data point is denoted by *d*, while *y* is the corresponding model prediction given the parameter vector $\vec{\theta}$. A weight factor $\gamma = 0.1$ was used in the case of insulin ($\gamma = 1$ in case of glucose) to account for the unit difference ($mmol/L$, $mU/L$ for glucose and insulin, respectively) between the molecules.

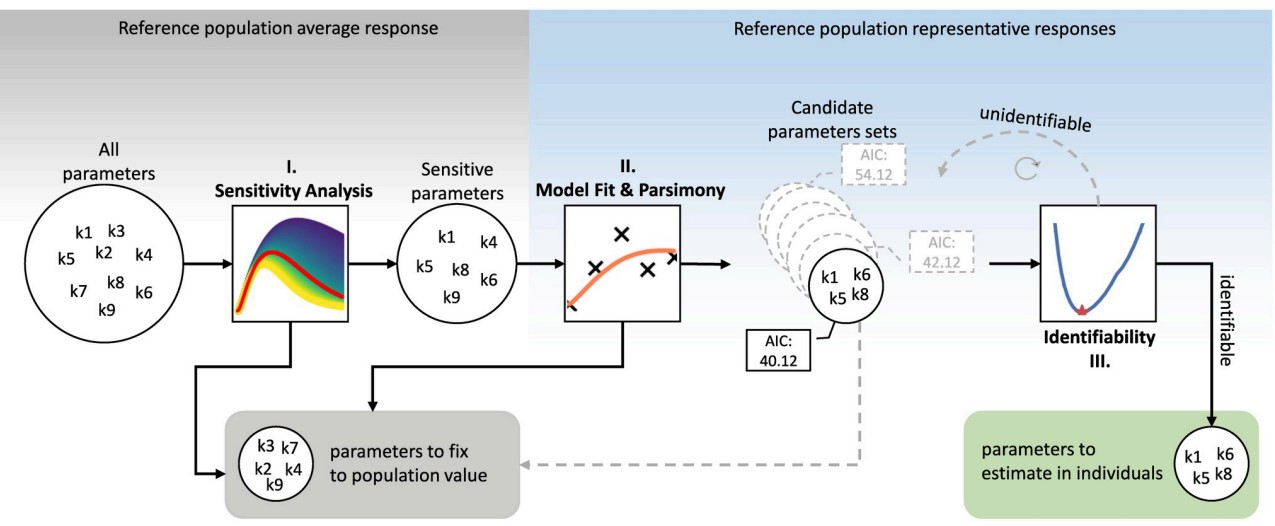

**Fig 1. Flowchart of the model selection approach.**

## Model selection

A model selection approach was implemented in order to systematically identify parameters for personalized model fitting. The aim of the approach is to maintain parameter functionality and identifiability when transitioning from modelling population average responses to individual responses. The workflow involves systematically reducing the number of parameters to estimate for each individual response to ensure reliable, accurate parameter estimates. The parameters that are not selected in the approach for personalized fitting are fixed to reference population values [24] across all individuals. The subset of model parameters to be estimated was selected based on the following criteria: the selected parameters had to (i) exhibit high sensitivity, (ii) demonstrate good model fit while maintaining parsimony, (iii) be identifiable. The steps to meet these criteria are detailed below and a flowchart of the approach is shown on Fig 1.

(i).  Sensitivity analysis. We carried out local parameter sensitivity analysis (LPSA) by varying one parameter in both directions while maintaining the others at a constant value, inspecting the effect on the resulting model outcome. A threshold of 75% in both directions compared to the average healthy population values was selected as the limit of sensitivity. Parameters that exhibit sensitivity at this level are expected to have a large modulatory effect on the model outcome. Parameters that are not sensitive at the threshold were fixed as a constant to their respective values (reported in [24]), as they have little impact on the model behavior. Only sensitive parameters were considered in subsequent steps of the model selection.

(ii).  Model fit. The set of all possible combinations of 3 or more sensitive parameters were generated. Subsequently models were fit on a set of representative responses from the DIOGenes data set with these candidate parameter sets estimated from the data, while the rest of the parameters were fixed to the population reference values. The representative responses comprised of the median normal glucose tolerant (NGT), impaired fasting glucose (IFG), impaired glucose tolerant (IGT), both IFG and IGT (IFG&IGT) and T2DM responses in the data, based on the American Diabetes Association (ADA) diagnosis criteria [29]. The median responses were calculated by taking the median glucose

and insulin values per time-point across all individuals in the respective groups. In addition to the median responses, both extreme responses (largest and smallest response in the data set by area under the glucose curve) were also included. The model with the candidate parameter set that showed the lowest Akaike Information Criterion (AIC) score across the set of representative curves (i.e. NGT, IFG, IGT, IFG&IGT, T2DM, Min, Max) was selected as most parsimonious model.

(iii).   Parameter identifiability. The parameter set that produced the most parsimonious model was finally evaluated for identifiability using Profile Likelihood Analysis (PLA) [27]. In PLA the value of one parameter is changed iteratively from its optimal value and the remaining parameters are re-estimated. An increase in the cost function (SSR) for the model fit indicates that a reliable parameter estimate has been obtained and the parameter is identifiable given the model structure and data. Confidence intervals were derived using a Chi-squared threshold on the likelihood (Eq 2).

$$-2log\left(\frac{\mathcal{L}(\vec{\theta}_{PL})}{\mathcal{L}(\vec{\theta}_{opt})}\right) \leq \chi^2(\alpha, df) \qquad (2)$$

Where $\chi^2(\alpha, df)$ is the $\alpha$ quantile of the $\chi^2$-distribution with $df$ degrees of freedom, $\vec{\theta}_{PL}$ and $\vec{\theta}_{opt}$ are the profiled path and optimal parameter vectors, respectively. The threshold $\alpha$ was set to 0.95 and $df$ equals one or the number of parameters (see S3 and S4 Figs)

## Principal component analysis

The parameter space of the personalized E-DES model is visualized by reducing the number of dimensions from the number of estimated parameters to two dimensions using principal component analysis (PCA). Prior to PCA, the parameter values were normalized to zero mean and unit standard deviation.

## Results

A total of 738 participants were included in the analysis, after excluding participants with incomplete OGTT measurements (n = 373) and participants with physiologically implausible responses (i.e. where the OGTT failed; n = 7). The identification of physiologically implausible responses was carried out by independent experts. The remaining 738 responses were characterized by the ADA criterion for prediabetes and diabetes as summarized in Table 1.

**Table 1. Classification of participant's responses based on ADA diabetes criteria.**

| Diagnosis[1] | NGT | IFG | IGT | IFG&IGT | T2DM |
|---|---|---|---|---|---|
| N | 496 | 42 | 41 | 119 | 40 |
| Age[2] [$years$] | 40.7 (6.4) | 42.0 (5.3) | 43.6 (4.8) | 41.8 (6.2) | 45.0 (6.7) |
| Sex [%$female$] | 65.8 | 42.9 | 53.7 | 68.1 | 55.0 |
| BMI[2] [$kg\,m^{-2}$] | 34.5 (4.8) | 34.7 (4.5) | 36.5 (5.8) | 34.2 (4.5) | 35.1 (5.1) |

[1] NGT: normal glucose tolerant, IFG: impaired fasting glucose, IGT: impaired glucose tolerant, T2DM: type 2 diabetes mellitus. For details about the criteria, see S1 Table.

[2] Age and BMI are reported as mean and (standard deviation).

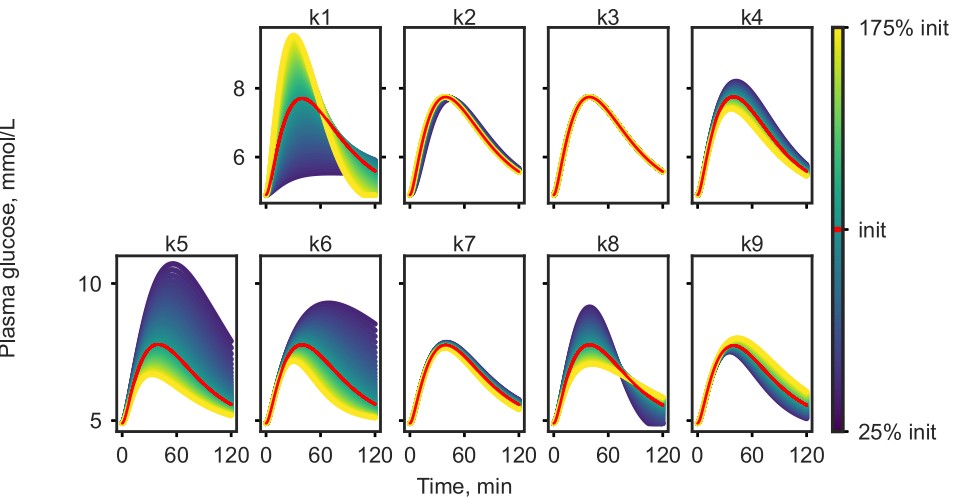

**Fig 2. Local parameter sensitivity analysis demonstrated on simulated plasma glucose response.**

In general, DIOGenes contains overweight/obese but otherwise healthy participants [28]. However, characterization by the ADA guidelines shows that in fact, several participants may be diagnosed as prediabetic or type 2 diabetic.

## Model selection

Out of the eleven parameters contained in the adjusted E-DES model, only the physiological parameters $k1$ to $k9$ (defined in Table C of S1 Appendix) were considered in the personalization, while the two remaining parameters, the shape factor and Michaelis-Menten constant for glucose uptake, were fixed to their respective population values. As the first step of the model selection approach local parameter sensitivity analysis was carried out. The outcome of the sensitivity analysis on the plasma glucose response is shown in Fig 2. The effect on plasma insulin can be seen in S2 Fig.

Parameters $k1$, $k4$, $k5$, $k6$, $k8$, $k9$ were found to be sensitive at the ±75% threshold, and therefore were considered for further analyses, while the remainder of the parameters were kept constant in all subsequent analysis. As the second step of the model selection, the set of all possible combinations of 3,4,5, and 6 sensitive parameters were generated and models with these parameters estimated from data were assessed. This way a total of 42 different models were examined for model fit according to AIC on the median NGT, IFG, IGT, IFG&IGT, and T2DM responses as well as the largest and smallest response in the data set. The ten best performing candidate models with the resulting SSR and AIC values are shown in S2 Table.

The highest scoring model according to our criteria contained the parameters $k1$, $k5$, $k6$, and $k8$ with a SSR of 41.39. Visual inspection of the model output displayed good accordance with the majority of the data on the various group median and extreme responses as seen in Fig 3. The extreme responses are simulated less accurately compared to the median responses. Specifically, the model struggles with accurately capturing the part of the response that goes below basal. The best scoring model was subsequently evaluated for parameter identifiability in the last step of the model selection approach.

The identifiability of the parameters $k1$, $k5$, $k6$, $k8$ was assessed on the median NGT, IFG, IGT, IFG&IGT, T2DM and the extreme responses to infer the reliability in estimating the selected parameters. The PLA profiles indicated that parameters were identifiable, with the exception of parameters $k6$ and $k8$, which were practically non-identifiable [27] for the lowest

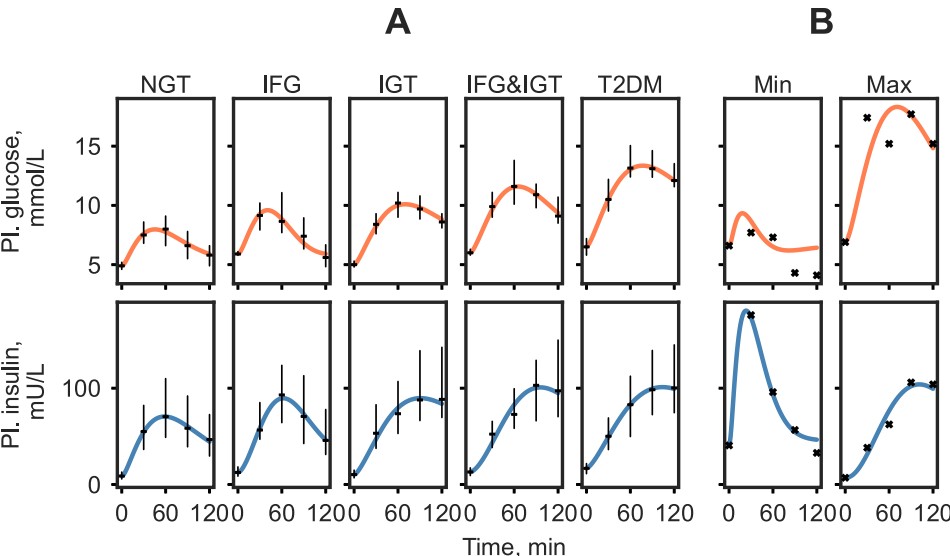

**Fig 3. Plasma glucose and insulin simulation of the set of representative responses in the DIOGenes study with estimated parameters $k1$, $k5$, $k6$, $k8$.** The median responses were calculated as the median plasma glucose value of the ADA diabetes classification group at each time point (A). The 'Min' and 'Max' are the smallest and largest glucose responses in the data set, determined by area under the curve (B). Median measured values are shown as black dashes with the interquartile range and measured responses are indicated by black crosses.

response. The parameter profiles can be found in S3 Fig. Additionally, to verify the choice of the 4 parameter model, the best performing 5 parameter candidate model (with parameters $k1$, $k5$, $k6$, $k8$, $k9$) was also evaluated for parameter identifiability. The PLA profiles from the 5 parameter model indicate that $k9$ was structurally non-identifiable in 6 out of 7 representative responses, with further two parameters ($k6$ and $k8$) proving to be non-identifiable in 2 out of 7 cases (S4 Fig).

The model selection pipeline resulted in a model with parameters $k1$, $k5$, $k6$, $k8$ to be estimated from experimental data in personalized models. The selected parameters describe the rate constant of glucose appearance in the gut ($k1$), the rate constant of insulin-dependent glucose uptake ($k5$), the proportional rate constant of insulin secretion due to the difference in the actual plasma glucose level compared to baseline ($k6$), and the insulin secretion dependent of the rate of change in plasma glucose ($k8$).

### Individual simulation

A population of 738 personalized models were generated *in silico* through estimating the selected parameters on post-load time series of glucose and insulin in participants from the DIOGenes study. To evaluate the success of simulating individual responses, we compared the discrepancy of population specific simulations to that of individualized simulations. The median response was calculated in every group (NGT, IFG, IGT, IFG&IGT, and T2DM) and the selected 4 parameter model was used to simulate the median glucose and insulin trajectories on the calculated responses. The individuals' measured data were then compared to the median simulations per group. The SSRs in the personalized model simulations were substantially lower than those of the median simulations in every group (Table 2), indicating that the personalized models were able to capture a wide range of response curves.

The best and worst personalized model simulations by SSR are shown in Fig 4A and 4B, respectively. While the measured glucose and insulin responses ranged from 1.8 to 18.3 mmol/

**Table 2. Mean (standard deviation) of sum of squared residuals in the model simulations.**

|  | NGT | IFG | IGT | IFG&IGT | T2DM |
|---|---|---|---|---|---|
| Group simulation | 149.87 | 213.52 | 205.15 | 153.46 | 195.93 |
|  | (153.11) | (205.15) | (134.92) | (126.97) | (181.90) |
| Individual simulation | 32.37 | 44.29 | 35.33 | 43.54 | 36.69 |
|  | (36.26) | (32.11) | (47.89) | (29.92) | (33.56) |

L and from 2.0 to 749.0 mU/L, the simulations show good agreement with the measured data in most cases. To highlight other striking model behavior, additional, hand selected example responses and their corresponding simulations are shown in Fig 4C.

In particular, metabolite responses with an intermediate dip between two values were found to be difficult to capture using the 4 parameter model (e.g. participant 183, Fig 4C).

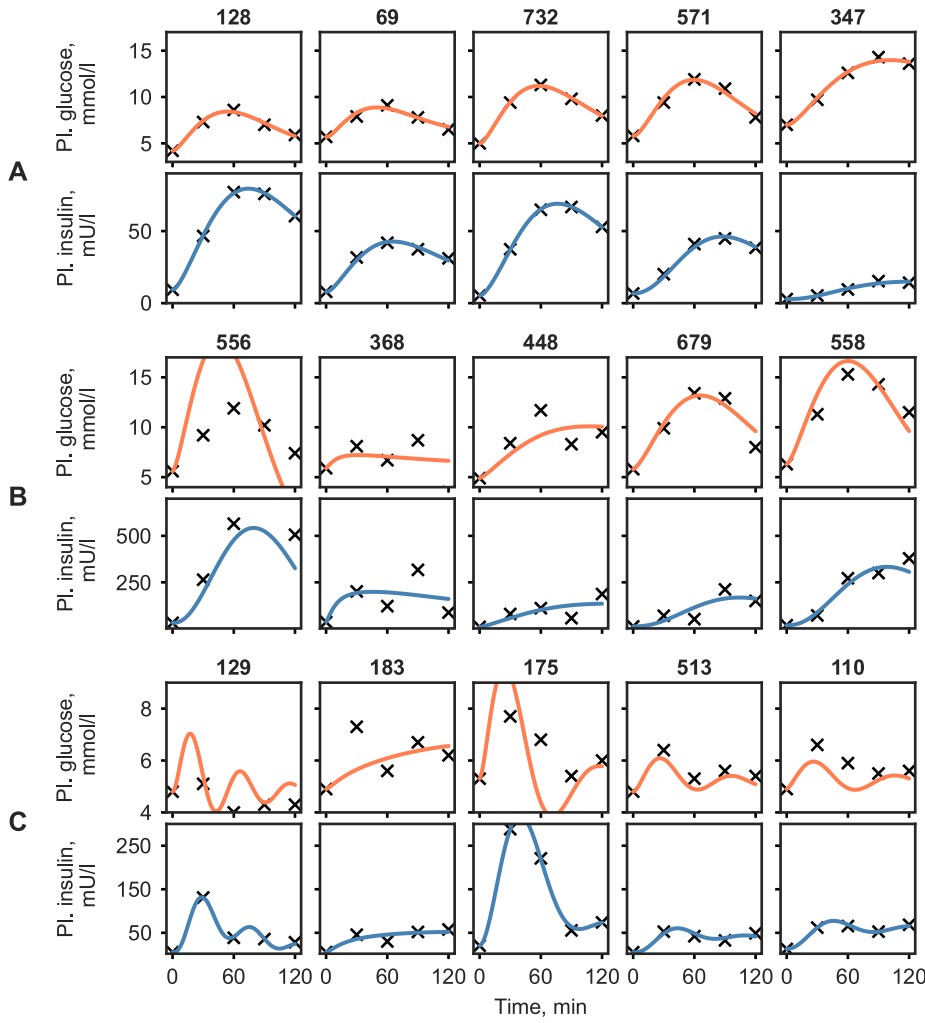

**Fig 4. Time courses of plasma glucose and insulin concentrations for individuals with the lowest and highest simulation error (quantified by the SSR) in each ADA category, and other interesting model behaviour (A, B and C respectively).** Black crosses and orange/blue lines correspond to measurement and model simulation of glucose/insulin, respectively.

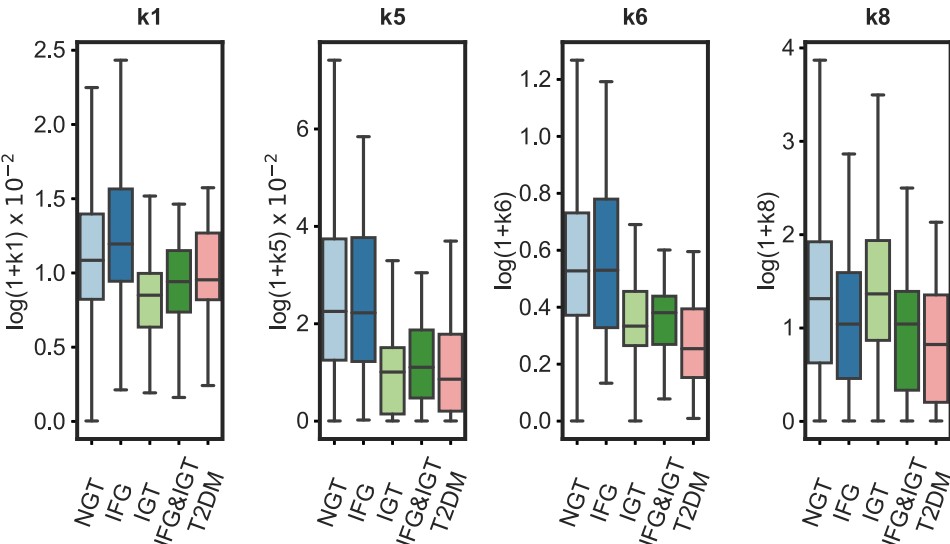

**Fig 5. The distribution of the estimated parameters *k*1, *k*5, *k*6, *k*8 by subgroup based on ADA diabetes criteria.**
*k*1—rate constant of glucose appearance in the gut, *k*5—rate constant of insulin-dependent glucose uptake, *k*6—rate constant of insulin secretion proportional to glucose elevation, *k*8—rate constant of insulin secretion by the rate of change in glucose elevation. For details, see S1 Appendix. Observations outside the interquartile range of the $25^{th}$ and $75^{th}$ percentile of each group are visualized in S5 Fig. The boxes represent the 25th and 75th percentiles, the whiskers represent the min, and max values, and the horizontal line represents the median.

However, there were cases of such bi-phasic curves, that the model could replicate accurately (e.g. participant 513, Fig 4C). In some cases, the model predicted a fast response, with a probable peak between the 0 and 30 minute measurements (e.g. participants 129, Fig 4C). Furthermore, the success of simulating complex shapes appeared to depend on the scale of the insulin values in the responses, where higher insulin values lead to difficulties in accurately fitting the glucose response (e.g. participant 175, Fig 4C).

The distribution of parameter values estimated from individuals' responses are shown by subgroup in Fig 5. In general, the range of estimated parameters was greatest in the group that was NGT according to the ADA diabetes criteria, with values spanning the whole range of the other groups. The rate constant of glucose appearance in the gut (k1) was largest in the NGT and IFG groups. Similarly, insulin-dependent glucose uptake (*k*5), and glucose-dependent insulin production (*k*6) were lower in the IGT, IFG&IGT, and T2DM groups compared to the NGT and the IFG groups. The plasma glucose rate of change-dependent insulin production (*k*8) was lower in the IFG&IGT, and T2DM groups compared to the other groups. Additionally, the association of the parameter values with frequently used measures of insulin secretion and insulin resistance were evaluated to assess model structure. Parameters *k*6 and *k*8 associated with the insulinogenic index ($r = 0.56$, $p < 0.001$ and $r = 0.49$, $p < 0.001$, respectively; S7 Fig), a frequently used measure of first-phase insulin secretion [30]. Additionally, parameter *k*5, describing insulin mediated uptake of glucose into the periphery showed a significant positive correlation with the Matsuda index (Pearson $r = 0.68$, $p < 0.001$; S7 Fig).

A better grasp of the parameter space of the model can be obtained by visualizing it after reducing the four dimensional space to two dimensions via principal component analysis. The personalized models in the resulting space are shown in Fig 6. The unique parameter set in each model defines the model's place in the parameter space, where the model is colored according to the ADA criteria (A) and the participants' Matsuda index (B). The explained

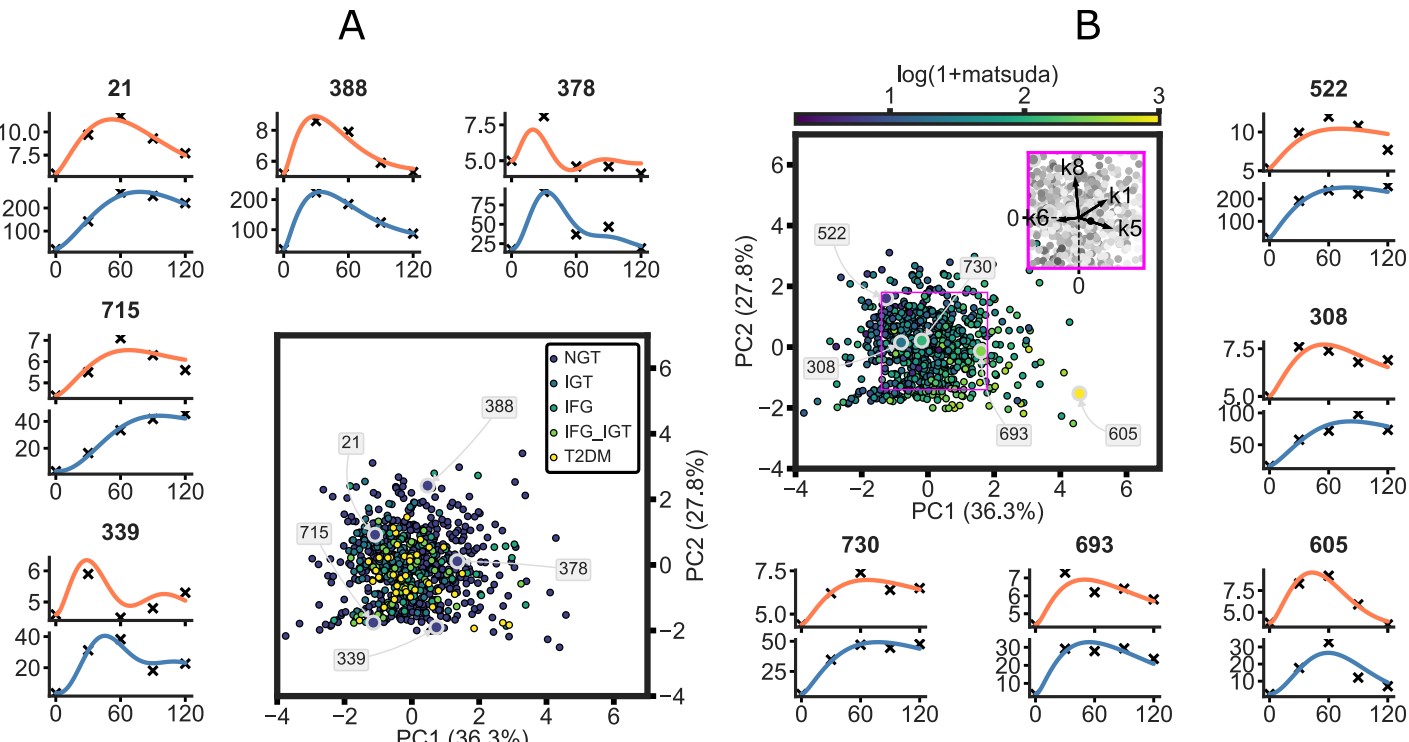

**Fig 6. Individual parameter sets in the parameter space of the model colored by the ADA diabetes criterion (A) and the Matsuda index (B) after reduction to 2d via principal component analysis.** The personalized model simulations of five participants' responses are highlighted (individuals 21, 388, 378, 715, 339 on panel A and 522, 308, 730, 693, 605 on panel B). Orange and blue lines correspond to glucose and insulin model simulation, while crosses represent measured data. The loading vectors of k1, k5, k6, and k8 are shown in the purple insert in panel B.

variance and the loading vectors indicate that the parameters pertain to distinct mechanisms and retain their functionality (Fig 6B insert). The spread over the first two principal components demonstrate the large heterogeneity in the modelled population. Furthermore, the additional insight gained by the personalized models is illustrated when coloring the parameter space by the ADA criterion for prediabetes and diabetes (Fig 6A). The featured examples highlight the large heterogeneity that remains hidden when considering only the single time-point measures of the diagnosis categories, but is captured by the person specific models. Fig 6B shows examples of participants with different states of insulin sensitivity. The examples positioned along the direction of the loading vectors of $k5$ show responses with increasing insulin sensitivity noted by the rapid clearance of glucose from plasma (i.e. curve rapidly approaching baseline) as well as lower insulin secretion. Responses of individuals with varying levels of first-phase insulin secretion are shown on S8 Fig with the corresponding models in the parameter space colored by the insulinogenic index. Responses along the loading vectors of both $k6$ and $k8$ indicate increasing first-phase secretion of insulin, with highlighted examples of low (e.g. participant 347) moderate (e.g. participant 693) and high (e.g. participant 51) secretion. Additionally, S9 Fig highlights examples in the parameter space colored by the error in the simulation of the individual as measured by SSR.

## Discussion

In this work, we implemented a pipeline to convert a physiological model of the postprandial glucose-insulin dynamics describing population averages into a personalizable model. A key

aspect of the parameter selection process was to maintain certainty in the parameter estimates and consequent model predictions by systematically reducing the number of parameters to be estimated in personalized models, taking into account the availability of quantitative data. Furthermore, our approach allows the comparison of personalized models due to retaining the same model structure across individuals. The protocol does not require biological knowledge and may be generalized to models of other systems. The resulting personalized models were able to accurately describe a wide variety of responses from the DIOGenes data set, a large population of overweight/obese but otherwise healthy individuals. Furthermore, the estimated parameter values of the model allowed mechanistic insight into the differences in individuals' glucose metabolism.

Parameters exhibiting high sensitivity exert a large effect on model outcome, whereas not sensitive parameters may be fixed to a constant value. Sensitivity of the parameters was assessed through their modulatory effect on glucose and insulin concentrations of an average healthy simulation, to keep in line with the study population. Due to the structure of the model, different responses might indicate different parameters to be sensitive, such as the parameter controlling the rate of endogenous glucose production, which is expected to behave differently when the glucose response goes below the basal level. However, in the current study population such behavior is rare (66/738 responses) and thus an average healthy simulation was considered adequate for sensitivity analysis. Following the selection of the sensitive parameters, parsimony and model fit was considered. The ADA group median and extreme responses were chosen to promote the model to be able to fit a wide range of responses. We hypothesize, that these responses are representative of the parameter space that we aim to capture with the model. Thus, if the model is able to capture these responses accurately, it is likely to be able to simulate arbitrary responses in intermediate states as well. In order to impose a criterion towards parsimony, we used the AIC to introduce a penalty term on the number of parameters in the model. As shown in S2 Table, candidate models with more parameters had a lower SSR. However, the top 5 and 6 parameter models performed only marginally better than the 4 parameter ones, with a SSR of 37.02, 35.20, 41.39 respectively. The best scoring 3 parameter model ($k1$, $k5$, $k9$) had more than twice the SSR (87.00) of the best 4 parameter model. Based on the AIC, the 4 parameter candidate model containing parameters $k1$, $k5$, $k6$ and $k8$ to be estimated was selected as the most parsimonious model. Finally, the identifiability of the candidate model was examined via PLA on the representative median and extreme responses. Besides the smallest response, PLA profiles showed that the parameters were identifiable (S3 Fig) indicating that a unique solution exists in the tested range.

The parameters identified in the model selection process indicate that the most discriminating processes in this population are transition of glucose from the stomach to the gut ($k1$), the insulin-dependent glucose uptake to the periphery ($k5$), and the processes representative of insulin secretion ($k6$ and $k8$). It is known that gastric emptying is a major determinant of postprandial glycemia that has been shown to exhibit large intra- and interindividual variability [31, 32]. The decline in insulin mediated glucose disposal into tissues such as the muscle, liver or adipose tissue is acknowledged as one of the key process leading to the development of T2DM [15]. Furthermore, defects in the first phase insulin secretion are known to appear in the early stages of deteriorating glucose control compared to abnormalities in second phase insulin secretion, which is more prevalent in advanced stages of T2DM [33]. The ability to potentially quantify these otherwise difficult to measure processes from time-series of postprandial glucose and insulin using a computation model may prove incredibly useful in the advent of personalized medicine and targeted nutritional interventions.

Following the identification of the model structure to be personalized, we elected to generate our population of personalized models by fitting the model to the individuals'

corresponding meal response data via maximum likelihood estimation (MLE) while fixing other parameters to population averages. Here, it is important to note that, approaches such as non-linear mixed effects modelling, where population and individual level dynamics are estimated simultaneously, may provide a valuable alternative to our approach [34, 35]. In addition, future applications may benefit from integrating regularization in the MLE as proposed by Dolejsch et al. [36].

The personalized models showed a 4-5 fold decrease in SSR in all groups compared to group simulations, confirming that the model personalization was successful, as well as re-enforcing the need for a personalized approach when assessing such dynamic responses. A good accordance with data was further confirmed by visually inspecting the model output (Fig 3 and S6 Fig). However, the SSR does not always give a realistic overview of the model fit, for instance, it can be susceptible to bias towards responses with extreme glucose and especially insulin values. Thus, to further highlight the limits of the model, a manual selection of responses and corresponding model simulations were shown in Fig 4C. The model frequently struggled with accurately predicting an intermediate dip in the glucose response (e.g. participant 183, Fig 4C). The more complex bi-phasic shapes were only accurately modelled in a few cases (e.g. participants 513, Fig 4C), although it is thought that, this lack of fit could be avoided by estimating additional parameters. Responses with high insulin values drove the model to fail at accurately capturing the glucose response. This is partly due to the combined glucose and insulin error function used in model fitting, in which insulin values were multiplied by 0.1 to account for the difference in scale compared to glucose. However, in case of extremely high insulin responses, the insulin values are still favored during the optimization (e.g. participants 175, 556, Fig 4B and 4C). By estimating additional glucose parameters, such as the parameter handling endogenous glucose production ($k3$), these responses might be captured more accurately. Additionally, in some cases where the glucose levels quickly returned below the basal value the model exhibited oscillatory behaviour (e.g. participants 129, Fig 4C). This may originate from the parameter estimates relating to the insulin secretion term in the model, however further examination of this was outside the scope of the current study. Furthermore, it is worth noting that the outlying parameter estimates not necessarily indicate erroneous simulations but rather unusual or extreme responses as can be seen on Fig 6, S8 and S9 Figs.

Each of the 738 personalized models contain a unique parameter set pertaining to the physiological state of the participant's glucose homeostasis. The largest range for all of the estimated parameters was found in the NGT group, which could partly be explained due to the data set containing more normo-glycemic individuals (see Table 1) resulting in a larger variability. Furthermore, normo-glycemic individuals are also known to be more likely to exhibit bi-phasic responses [11], raising the variability of responses, and thereby the range of estimated parameters values in this category. In addition, the groupings defined by the ADA criteria only consider the fasting and 2h plasma glucose values while ignoring the insulin levels. Thus, individuals that exhibit normal glucose levels at the fasting and 2h time-points due to unusually high insulin values still end up in the NGT group. This lack of consideration for the dynamics and insulin values make it difficult to detect early deterioration in individuals' responses indicative of insulin resistance. However, taking into account the complete dynamics of both glucose and insulin the personalized models outlined here are able to indicate such transitions before they are detected by steady state or single time-point measures (e.g. 221, 522 Fig 6). By screening for the parameter estimates of k6 and k8 one can identify cases where the glucose response appears normoglycemic, however the insulin levels are abnormally high.

Variation in gastric emptying linked to obesity has been previously reported, however we found no difference in the parameter estimates for $k1$ between overweight and obese participants as well as no association between the parameter $k1$ and BMI [37]. Importantly, the model parameters corresponding to insulin secretion ($k6$, $k8$) were found to be lower for individuals with more severe metabolic conditions (IGT, IFG&IGT, T2DM). The insulin secretion parameters also showed a significant association with the insulinogenic index, a frequently used measure of insulin secretion. While the parameter controlling the insulin-dependent glucose uptake $k5$ was lower in IGT, IFG&IGT and T2DM compared to the other groups and showed a significant association with the Matsuda index. These findings reinforce that the model structure captures relevant features of the insulin mediated glucose homeostasis and the personalized models can distinguish between divergent impairments in the insulin regulated glucose control. Therefore, our modeling framework might prove beneficial in revealing nuanced behaviour specifically for the early detection of decline in the glucose homeostasis from a standard five time-point OGTT. Moreover, the personalized models may be used to assess the effects of lifestyle and diet interventions, where the observed effects can be quite subtle. Our results also highlights the possibility of using such an approach to generate cohorts of virtual patients with varying glucose homeostasis for potential *in silico* testing.

The population in the study may be considered relatively homogeneous in terms of glucose homeostasis, as measured by current single time-point measures such as the ADA criterion. However, the personalized models utilizing the dynamic, intertwined plasma glucose and insulin responses of individuals, allowed the quantification of an immense heterogeneity in the responses even within the ADA groups. Furthermore, the mechanistic nature of the model promotes the identification and allows comparison of distinctive processes underlying individuals' metabolic health. We believe that such personalized modelling approaches will be essential in advancing personalized nutrition.

## Conclusion

The systematic model selection pipeline implemented in this work allows the personalization of a mathematical model through reducing the number of parameters to be estimated in personalized models. The approach results in the most parsimonious model that contains identifiable parameters. The selection pipeline is generalizable in the sense, that it does not require biological insight to implement, therefore it may be applied to other systems or models to gain insight on the individual level. The E-DES model, a computational model of the human glucose-insulin system, was personalized using the approach and subsequently a population of personalized models were simulated from a large data set of overweight/obese but otherwise healthy individuals. The personalized models, consisting of only four parameters estimated from experimental data were capable of simulating a wide variety of postprandial glucose and insulin responses to a standard OGTT from the DIOGenes data set. Taking advantage of a frequently sampled time-series of both glucose and insulin the dynamic models were able to capture a large, previously overlooked heterogeneity in the population. The mechanistic aspect of the model allows the description and comparison of the physiological state of the individuals' glucose homeostasis and provide mechanistic insight into the glycemic variability observed in the responses.

## Supporting information

**S1 Fig. Schematic of the E-DES model in use.**
(TIF)

**S2 Fig. Local parameter sensitivity analysis on the simulated plasma insulin response.** (TIFF)

**S3 Fig. Profile Likelihood Analysis results of the 4 parameter model ($k1$, $k5$, $k6$, $k8$) on the median NGT, IFG, IGT, IFG&IGT, T2DM, min and max responses.** The red star indicates the SSR of the model fitted using the optimal parameter values estimated from data, while the blue line corresponds to the error as the other parameter values are being re-estimated after adjusting the parameter value iteratively. The dashed lines indicate confidence intervals where the degrees of freedom equals one (lower), and the number of parameters (upper), respectively. (TIF)

**S4 Fig. Profile Likelihood Analysis results of the 5 parameter model ($k1$, $k5$, $k6$, $k8$, $k9$) on the median NGT, IFG, IGT, IFG&IGT, T2DM, min and max responses.** The red star indicates the SSR of the model fitted using the optimal parameter values estimated from data, while the blue line corresponds to the error as the other parameter values are being re-estimated after adjusting the parameter value iteratively. The dashed lines indicate confidence intervals where the degrees of freedom equals one (lower), and the number of parameters (upper), respectively. (TIF)

**S5 Fig. The distribution of parameters $k1$, $k5$, $k6$, $k8$ by subgroup (based on ADA diabetes criteria) with the outliers highlighted.** The boxes represent the 25th and 75th percentiles, the whiskers represent the min, and max values, and the horizontal line represents the median. (TIFF)

**S6 Fig. Pooled residuals in the personalized models per time-point per metabolite, colored by the ADA prediabetes and diabetes diagnosis criteria.** (TIFF)

**S7 Fig. Pairwise scatter plots and density plots of the personalized model parameters, the Insulinogenic index and the Matsuda index from the DIOGenes study colored by the ADA prediabetes and diabetes diagnosis criteria.** (TIF)

**S8 Fig. Personalized models colored by the Insulinogenic index in the parameter space of the model after reduction to 2d via principal component analysis.** The personalized model simulations of five participants with varying first-phase insulin secretion are highlighted (individuals 51, 45, 347, 430, 693). Orange and blue lines correspond to plasma glucose and insulin model simulation, while crosses represent measured data. (TIFF)

**S9 Fig. Personalized models colored by SSR in the parameter space of the model after reduction to 2d via principal component analysis.** The personalized model simulations of five participants with varying first-phase insulin secretion are highlighted (individuals 175, 738, 556, 676, 445). Orange and blue lines correspond to plasma glucose and insulin model simulation, while crosses represent measured data. (TIFF)

**S1 Appendix. E-DES model structure, fluxes, inputs, parameters and constants.** (PDF)

**S2 Appendix. MATLAB implementation of the model used in the manuscript.** (ZIP)

**S1 Table. Criteria for prediabetes and diabetes classification used in this study.** Based on the standard two hour OGTT by the American Diabetes Association.
(PDF)

**S2 Table. Results of step two of the model selection approach.** Sum of squared residuals (SSR) and Akaike Information Criterion (AIC) of the ten best performing candidate models.
(PDF)

## Acknowledgments

The authors would like to thank the staff and participants of the DIOGenes study.

## Author Contributions

**Conceptualization:** Balázs Erdős, Bart van Sloun, Ilja C. W. Arts, Natal A. W. van Riel.

**Data curation:** Balázs Erdős, Bart van Sloun, Dominique Langin, Arne Astrup, Ellen E. Blaak.

**Funding acquisition:** Michiel E. Adriaens, Ellen E. Blaak, Ilja C. W. Arts, Natal A. W. van Riel.

**Investigation:** Balázs Erdős, Bart van Sloun, Ilja C. W. Arts.

**Methodology:** Balázs Erdős, Bart van Sloun, Shauna D. O'Donovan, Natal A. W. van Riel.

**Project administration:** Balázs Erdős, Ellen E. Blaak.

**Resources:** Michiel E. Adriaens, Dominique Langin, Arne Astrup, Ellen E. Blaak.

**Software:** Natal A. W. van Riel.

**Supervision:** Michiel E. Adriaens, Ilja C. W. Arts, Natal A. W. van Riel.

**Validation:** Shauna D. O'Donovan.

**Visualization:** Balázs Erdős, Michiel E. Adriaens, Shauna D. O'Donovan.

**Writing – original draft:** Balázs Erdős, Bart van Sloun.

**Writing – review & editing:** Balázs Erdős, Bart van Sloun, Michiel E. Adriaens, Shauna D. O'Donovan, Dominique Langin, Arne Astrup, Ellen E. Blaak, Ilja C. W. Arts, Natal A. W. van Riel.

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
