## [Decision Letter · Decision Letter 0]

14 Dec 2020

Dear Mr. Erdos,

Thank you very much for submitting your manuscript "Individualized computational model quantifies heterogeneity in postprandial responses to oral glucose challenge" for consideration at PLOS Computational Biology.

As with all papers reviewed by the journal, your manuscript was reviewed by members of the editorial board and by several independent reviewers. In light of the reviews (below this email), we would like to invite the resubmission of a significantly-revised version that takes into account the reviewers' comments.

We cannot make any decision about publication until we have seen the revised manuscript and your response to the reviewers' comments. Your revised manuscript is also likely to be sent to reviewers for further evaluation.

Sincerely,

James Gallo

Associate Editor

PLOS Computational Biology

Jason Papin

Editor-in-Chief

PLOS Computational Biology

Reviewer's Responses to Questions

**Comments to the Authors:**

Reviewer #1: Reproducibility report has been uploaded as an attachment.

Reviewer #2: Comments for the Author

The manuscript by Erdos B. et al. simulates the data of oral glucose tolerance test for overweight / obese participants in DIOGene study. The authors previously launched E-DES model using already existing dataset of OGTT and in this research by fitting to individual data the weight of each parameter and its correlation to physiological findings. This model seems to be useful for estimating mechanistic insight such as individual insulin resistance or endogenous insulin secretion and evaluating the effect of interventions.

1 In previous paper (Mass AH, et al. 2015), the parameters k5, k6, k7 and k8 were the most sensitive to change by a multiparametric sensitivity analysis. In this research, the parameter k7 is omitted on the basis of LPSA while the parameter k1 is employed to following analysis.

1.1 What is the reason of these discrepancies between previous research and this research? Does it reflect the difference in backgrounds between previously used dataset and the data of DIOGenes study?

1.2 The parameter k1 changes the peak concentration and time of gut glucose mass resulting in plasma glucose concentration. The authors should discuss about the association between the factor regulating parameter k1 (e.g. intestinal motility) and the variation of overweight / obesity subjects.

1.3 Is the parameter k1 also higher level in NGT or IFG than ones in IGT, IFG&IGT and T2DM? As well as the parameters k5, k6 and k8 (line 250-254), this should be mentioned.

2 Although the authors classify the participant in DIOGenes according to ADA diagnosis criteria (line 144-147), the median of blood glucose in figure 2 seems strange. Since there may be some errors in column annotations, please explain with actual median values and correct if needed. When corrected, please mention about the rightfulness of following analysis.

2.1 Plasma glucose level at 120 min in IGT seems lower than 140mg/dL.

2.2 Plasma glucose level at 120 min in IFG seems higher than 140mg/dL.

2.3 Plasma glucose levels at 0 min in IGT and IGF are higher than one in IFG&IGT.

3 Can this model predict the OGTT patterns of new subjects? In other words, are these significant parameters of k1, k5, k6 and k8 able to be predicted by some other physical parameters (e.g. Body mass index, HOMA-IR / Matsuda index, HbA1c, etc)?

4 PCA analysis for these four parameters shown in figure 5 is very interesting. At a glance, NGT is a most heterogenous population but no significant difference between groups (e.g. no group make clear cluster or tendency). In other words, no differential tendency of diagnostic classification along with vectors of the parameter k1, k5, k6 and k8 is shown. In terms of the pathogenesis of diabetes, T2DM will show lower insulin secretion (k6 and k8) and/or insulin resistance (k5). How the authors explain this discrepancy? The opposite direction of k5 and one of k6 or k8 may be explanation for it, is it a feature of this modeling or used data set?

5 In figure S5, the association between SSR and the four parameters are well described. Participant 175 shows higher SSR resulting incorrect simulation for the parameters. While participant 556 shows notably high SSR but the parameters locates within an average range. This may be a result of the difference in absolute value of plasma glucose concentration. The authors may be better to normalize the SSR using absolute value.

6 The parameters visualized to reflect the physical status are most important and interesting in this research.

6.1 The authors visualized the correlations between these parameters and insulin sensitivity in figure 5. While endogenous insulin secretion from pancreatic beta cells is also important on the pathogenesis of type 2 diabetes. Insulinogenic index calculated from OGTT result is often good indicator of endogenous insulin secretion. The association between these parameters and the indicator of endogenous insulin secretion should be shown.

6.2 In this study, can any alteration in these parameters after intervention be observed?

Reviewer #3: In this manuscript, the authors fitted their previous well-established physiology-based dynamics model of glucose-insulin system to individuals' data to elucidate the heterogeneity in individuals' responses. In order to reduce the number of estimated parameters for individualized models, the authors did sensitivity analysis, model selection based on Akaike information criterion (AIC), examined the parameter identifiability using profile likelihood, and finally chose the model with four parameters (k1, k5, k6, k8) to be estimated. The individualized model fitted pretty good, although the authors also reported its inability in some cases. Based on the estimated parameters of individualized models, the authors also did principal component analysis (PCA) and showed correlation between k5 (rate constant of insulin-dependent glucose uptake) with Matsuda index.

Overall, this is a pretty good study. The manuscript looks quite clear and easy for understanding. The purpose of this study developing individualized models seems to elucidate the heterogeneity in individuals' responses. It might be better if the authors could provide some more details in the application of these individualized models. And I listed some other comments as follows.

Major comments:

1. Page 10/22 Line 200, 'inability to go below fasting value', is it possible that this behavior is due to magnitude difference in k10 and k3 (as shown in Appendix S1 Table 3). In fact, I did not find k10 = 2.60 in ref [23]; I am not sure if I missed something.

With similar thoughts, since these rate constants k1-k10 are all multiplicative in the dynamics equations, maybe the authors should explore some magnitude of possible values (such as 0.01X, 0.02X, 0.05X, 0.1X, ... to 100X) instead of current explred range 25% to 175%? And maybe that is part of the reason why some of the rate constants are not sensitive (as shown in Fig 1).

2. Page 12/22 Line 255, 'significant positive correlation with the Matsuda index'. Could the authors show a scatter plot between the two variables for the reported correlation?

Similarly, for better understanding of the heterogeneity, it might be better to also show the pairwise scatter plot with density contour between k1, k5, k6 and k8, in additional to the boxplot in Fig 4 and the PCA plot in Fig 5.

3. Page 14/22 Line 318, 'as this type of reponse is rare in modelled population'. How many cases are with this type of response? Will it help if the authors first select out these cases, and then estimate the parameters on this group of people?

4. Page 15/22 Line 335, 'The individualized models showed a 4-5 fold derease in SSR in all groups compared to group simulations, indicating that the model individualization was successful'. I am wondering the individualized model is estimating 4 parameters to fit 8 data points, right? It might be expected that individualized models would have smaller SSR. Although the authors have shown some cases of under-fitting (Fig 3-B, 3-C, and Fig S5), is it possible for the individualized models over-fitting the data? In other words, how reliable are the estimated parameters of individualized models?

If it is estimating 4 parameters to fit 8 data points, and over-fitting might be an issue, then two possible suggestions might be: (1) using Bayesian framework on parameter estimation; (2) exploring the relationship between individuals by similarity in their data points (e.g. hierarchical clustering or tSNE embedding), and then estimating and possibly smoothing the paramters considring the group of neighbors.

5. Fig 3B, participant 556, it looks weird to see a sharp angle on the right of the red curve when it reaches the fasting level, which might indicate that the parameters for glucose level less than its fasting level are wrongly specified in magnitude difference, which might be the reason for the inability of the model going below fasting value.

Minor comments:

1. Page 6/22 Line 118, 'a weight of 0.1', is the weight inside or outside the square of formula (1)? And m and N in formula (1) might be better defined or explained.

Instead of fitting to median (or min or max) responses, is it possible or better to estimate the parameters with confidence interval of the model by fitting it to all the data for each group (NGT, IGF, IGT, IGF&IGT, and T2DM)?

2. Page 7/22 Line 132, 'original publication' might be better have the citation to ref [23].

3. Page 8/22 Line 152, 'seven curves' looks unclear and confuses me, until I saw Fig 2.

4. Page 8/22 Line 161, 'X^2(alpha, df)', what number are used as alpha and df? And this term 'X^2(alpha, df)' is not shown exactly the same in formula (2).

5. Page 11/22 Line 241, 'the model could replicate accurately (e.g. participants 110, 513, Fig 3-C).' The fitting to data points of participant 110 looks not so accurate, while the fitting to data points of participant 513 looks much more accurate.

6. Fig 4 legend, box-plot elements might be better defined, such as median, box limits, and whiskers. And 'Fig S3' in Fig 4 legend might be typo of 'Fig S4'.

7. Page 16/22 Line 367, 'individuals ... due to unually high insulin values still end up in the NGT group'. How many such cases in 496 NGT individuals?

8. Fig 1, the label of y-axis might be better as 'Plasma glucose, mmol/L'; or mention it in figure legend.

9. Fig 5A-B, the color coding might be too close for distinguishing (e.g. IGT vs IFG); and for matsuda index, it might be better colored in monocolor, e.g. white-green-black.

10. Fig 5B, The loading vector subplot is not mentioned in figure legend; and no explanation of the color of the gray scatter points in that subplot.

11. Table S2, it might be better if the authors could show some percentiles (e.g. 5%, 25%, 50%, 75%, 95%) in addition to the mean and standard deviation of each parameter. And since the parameters are multiplicative, and Fig 4 shows near normal after log-transformation, the standard deviation without log-transformation might not be very meaningful.

In addition, it might be helpful if the authors could compare Table S2 with the estimates with confidence intervals of the models fitted to the median responses.

Reviewer #4: Uploaded as an attachment.

**Have all data underlying the figures and results presented in the manuscript been provided?**

Reviewer #1: None

Reviewer #2: Yes

Reviewer #3: **No: **The authors stated that the original data are not publicly available due to ethical restrictions and privacy, but are available on request from the DIOGenes study data manager. The MATLAB code has been provided as Appendix S2.

Reviewer #4: Yes

PLOS authors have the option to publish the peer review history of their article (what does this mean?). If published, this will include your full peer review and any attached files.

Reviewer #1: No

Reviewer #2: No

Reviewer #3: No

Reviewer #4: **Yes: **Lindsay Clegg
---

## [Decision Letter · Decision Letter 1]

3 Mar 2021

Dear Mr. Erdos,

We are pleased to inform you that your manuscript 'Personalized computational model quantifies heterogeneity in postprandial responses to oral glucose challenge' has been provisionally accepted for publication in PLOS Computational Biology.

Best regards,

James Gallo

Associate Editor

PLOS Computational Biology

Jason Papin

Editor-in-Chief

PLOS Computational Biology

Reviewer's Responses to Questions

**Comments to the Authors:**

Reviewer #1: The reproducibility report has been uploaded as an attachment.

Reviewer #2: The manuscript has been revised satisfactory.

Reviewer #3: The authors have made great effort to improve their manuscript, and have addressed all my concerns.

**Have all data underlying the figures and results presented in the manuscript been provided?**

Reviewer #1: None

Reviewer #2: None

Reviewer #3: Yes

PLOS authors have the option to publish the peer review history of their article (what does this mean?). If published, this will include your full peer review and any attached files.

Reviewer #1: **Yes: **Anand K. Rampadarath

Reviewer #2: No

Reviewer #3: **Yes: **Adam Yongxin Ye

---

## [Editor Report · Acceptance letter]

16 Mar 2021

PCOMPBIOL-D-20-01381R1 

Personalized computational model quantifies heterogeneity in postprandial responses to oral glucose challenge

Dear Dr Erdos,

I am pleased to inform you that your manuscript has been formally accepted for publication in PLOS Computational Biology. Your manuscript is now with our production department and you will be notified of the publication date in due course.

With kind regards,

Alice Ellingham
